# α-Lack-SPI Alleviates MASLD in Rats via Regulating Hepatic Lipid Accumulation and Inflammation

**DOI:** 10.3390/nu17182918

**Published:** 2025-09-10

**Authors:** Mingtao Chen, Shanshan Guo, Xuye Lai, Qiyao Xiao, Xueqian Wu, Jinzhu Pang, Lei Pei, Yingying Gu, Xuguang Zhang, Lili Yang

**Affiliations:** 1Guangdong Provincial Key Laboratory of Food, Nutrition and Health, Department of Nutrition, School of Public Health, Sun Yat-sen University, Guangzhou 510080, China; chenmt39@mail2.sysu.edu.cn (M.C.); laixy23@mail2.sysu.edu.cn (X.L.); xiaoqy27@mail2.sysu.edu.cn (Q.X.); wuxq65@mail2.sysu.edu.cn (X.W.); peilei3@mail2.sysu.edu.cn (L.P.); guyy23@mail2.sysu.edu.cn (Y.G.); 2Mengniu Institute of Nutrition Science, Global R&D Innovation Center, Inner Mongolia Mengniu Dairy (Group) Co., Ltd., Hohhot 011050, China; guoshanshan@mengniu.cn (S.G.); pangjinzhu@mengniu.cn (J.P.); fxgzhang@mengniu.cn (X.Z.)

**Keywords:** MASLD, α-lack-SPI, high-fat, high-cholesterol diet, hepatic lipid metabolism, inflammation

## Abstract

**Background:** Metabolic Dysfunction-Associated Steatotic Liver Disease (MASLD) has become a worldwide health concern. Soy protein isolate (SPI) is a plant-derived protein with high nutritional value and has shown promising effects in regulating lipid metabolism and inflammation. **Objectives:** This study aimed to investigate the effects of an α-subunit-deficient SPI (α-lack-SPI) on MASLD and the underlying molecular mechanisms. **Methods:** Rats were fed with a high-fat, high-cholesterol diet (HFD) to induce MASLD. **Results:** The results showed that α-lack-SPI significantly reduced the levels of hepatic TG and TC, serum ALT, AST, TC, and LDL-C, and increased serum HDL-C in rats with HFD-induced MASLD. α-lack-SPI significantly attenuated hepatic steatosis and hepatocyte ballooning revealed by histopathological analysis. Meanwhile, α-lack-SPI markedly downregulated the mRNA expressions of Srebf1, Acaca, Fasn, Pcsk9, and Hmgcr, while significantly upregulating Pparα. Additionally, α-lack-SPI treatment significantly reduced the mRNA expressions of hepatic pro-inflammatory cytokines (Tnf-α, Il-1β, Il6), chemokine (Ccl2), and inflammasome component (Nlrp3), as well as the protein expression of COX-2. **Conclusions:** In conclusion, α-lack-SPI alleviated MASLD in HFD-fed rats probably via improving hepatic lipid metabolism and mitigating hepatic inflammation. These findings indicate that α-lack-SPI may serve as a promising nutritional intervention for MASLD management.

## 1. Introduction

MASLD, the most prevalent chronic liver disease worldwide, is characterized by excessive hepatic lipid accumulation driven by systemic metabolic dysfunction [1]. MASLD has been estimated to affect 30% of the adult population worldwide during 1999–2019, and is becoming an important public health problem all over the world [2,3]. Metabolic dysfunction-associated steatohepatitis (MASH) is a progressive type of MASLD, histologically characterized by lobular inflammation and hepatocyte ballooning, and is associated with a greater risk of fibrosis progression [4]. Currently, MASLD’s high prevalence and mortality impose a significant economic burden globally [5].

Hepatic lipid accumulation and liver steatosis are recognized as the primary trigger of MASLD pathogenesis, with de novo lipogenesis (DNL) serving as a central driver [6]. In patients with MASLD, the expression levels of the two crucial enzymes—acetyl-CoA carboxylase (ACC) and fatty acid synthase (FAS)—are markedly elevated, along with a robust increase in the master regulator sterol regulatory element-binding protein 1c (SREBP-1c), all of which are involved in the DNL pathway [7,8,9]. The resulting lipid overload subsequently provokes lipotoxicity, initiating and perpetuating chronic hepatic inflammation that fuels progression to MASH [10,11]. Moreover, systemic cholesterol homeostasis is intimately linked to the progression of MASLD, and elevated lipid levels are frequently observed in MASLD patients [12,13].

Recently, the first MASH treatment drug against liver fibrosis was approved by the US Food and Drug Administration [14]. However, no drug has been formally approved for use in patients with MASLD. Thus, there is an urgent need for developing novel measures for MASLD/MASH treatment. Dietary supplements with antioxidant, anti-inflammatory, and metabolic benefits have emerged as promising adjuncts to mitigate liver fat, inflammation, and slow down the progression of MASLD and MASH [15].

Driven by recent advances in food science and processing technology, soy protein isolate (SPI) has been widely adopted as a functional ingredient to boost the nutritional value of processed foods [16,17]. SPI is mainly composed of two principal classes of proteins, β-conglycinin (7S globulin) and glycinin (11S globulin), which constitute approximately 70% of the total protein content. To be specific, 7S is a trimeric glycoprotein (150–200 kDa), mainly comprising α (72 kDa), α′ (76 kDa) and β (53 kDa) peptides, while 11S globulin is a hexamer comprising five subunits, each comprising an acidic subunit A (35 kDa) and basic subunit B (20 kDa), which are linked by disulfide bonds [18]. Previous studies have found that SPI possesses the capacity to ameliorate lipid profiles, exert antioxidant effects, and modulate inflammatory cytokines [19]. Moreover, SPI exerted a protective effect against hepatic steatosis by reducing inflammation and enhancing lipid export from the liver in obese Zucker rats [20,21,22]. Extensive evidence indicates that SPI robustly modulates lipid metabolism, with distinct subunit compositions conferring differential regulatory capacities [18,23].

Currently, the nutritional benefits of SPI composed of different subunits are being extensively investigated [24,25,26]. Studies have shown that the absence of the α subunit in 7S globulin leads to a compensatory increase in the α′ subunit [23]. The α′ subunit, which plays a major role in the LDL-C-lowering properties of the 7S globulin, increases hepatic LDL uptake, ameliorating hypercholesterolemic conditions and concomitantly reducing serum TG [27,28]. Additionally, researchers have developed a transgenic rice with cholesterol-lowering activity by utilizing the α′ subunit of 7S globulin [29]. Recent studies have indicated that SPI lacking the α-subunit holds potential capabilities in glycolipid metabolism [18]. However, the role of SPI lacking the α-subunit in the prevention and treatment of MASLD remains largely unexplored, with its underlying mechanisms yet to be elucidated, warranting further investigation.

In this study, we investigated the therapeutic potential and underlying mechanisms of a specialized soy protein isolate α-lack-SPI in a high-fat, high-cholesterol diet-induced MASLD rat model, focusing on its effects on regulating hepatic lipid metabolism and inhibiting inflammation.

## 2. Materials and Methods

### 2.1. Materials

The wild-type soy protein isolate (WT-SPI) and α-subunit-deficient soy protein isolate (α-lack-SPI) were provided by Shandong Yuxin Bio-Tech Co., Ltd. (Qingdao, China) The subunit compositions of these two SPIs are shown in Appendix A. The SPF grade control chow control diet (CON) (Catalog No.: GDMLAC-260) and high-fat, high-cholesterol chow diet (HFD) (Catalog No.: GDMLAC-102, Appendix A) were purchased from the Guangdong Medical Laboratory Animal Center (Foshan, China). The high-fat, high-cholesterol diet was comprised of 52.2% basic feed, 20.0% sucrose, 15.0% lard, 10.0% casein, 1.2% cholesterol, 0.2%sodium cholate, 0.6% calcium hydrophosphate, 0.4% premixed feed, and 0.4% mountain flour.

### 2.2. Animal Experiments

Thirty-two male Sprague–Dawley rats (5 weeks old, 180–240 g) were purchased from the Experimental Animal Center of Sun Yat-sen University (Guangzhou, China, License No.: SCXK (Yue) 2021–0029). All animal experiments were approved by the Animal Ethics Committee of Sun Yat-sen University (2024001638). The rats were housed in a specific pathogen-free (SPF) environment with the temperature maintained at 25 °C, humidity at 55%, and a 12-h light-dark cycle. Rats were provided with ad libitum access to food and water throughout the study period and were permitted to acclimate to the new environment for 1 week prior to the commencement of experiments.

After a one-week acclimation period, rats were randomly assigned to two groups: the control chow diet group (CON, *n* = 8) and the high-fat, high-cholesterol diet group (HFD, *n* = 24). Rats in the CON group were fed a control chow diet, continued until the end of the experiment, while those in the HFD group were fed a high-fat, high-cholesterol diet. After 4 weeks, the HFD group was randomly divided into three experimental groups: HFD, HFD + WT-SPI (730 mg/kg/day), and HFD + α-lack-SPI (730 mg/kg/day). Suspensions of each compound were prepared by dispersing them in 5 mL of 0.05% carboxymethyl cellulose sodium solution. These suspensions were administered daily via gavage. Rats in the CON and HFD groups received an equivalent volume of the vehicle alone, also based on their body weight, to the treatment group. The health status of the rats was monitored daily, and the food intake was recorded. The body weight of the rats was measured every three days. The treatment lasted for a total of 34 days, after which euthanasia was performed on rats through cardiac puncture under isoflurane anesthesia. Blood and liver tissue were then collected for further analysis. A flow diagram for the study is shown in Figure 1A.

### 2.3. Measurements of Serum and Liver Biochemical Markers

Serum from rats was collected by centrifugation at 2500× *g* for 10 min at 4 °C. The serum samples were stored at −80 °C. Serum levels of total cholesterol (TC), TG, LDL-C, high-density lipoprotein cholesterol (HDL-C), alanine transaminase (ALT), and aspartate transaminase (AST) were measured using commercial assay kits purchased from Jiancheng Bioengineering Institute (A111-1-1, A110-1-1, A113-1-1, A112-1-1, C009-2, and C010-2, Nanjing, China). Liver TG and TC were measured with kits from Applygen Technologies Inc. (E1013 and E1015, Beijing, China). Measurements were performed using a Spark 10M multimode microplate reader (Tecan Trading AG, Männedorf, Switzerland).

### 2.4. Histopathological Analysis

For histological analysis, rat liver tissue samples were fixed in 4% paraformaldehyde solution overnight, dehydrated using organic reagents. The samples were subsequently embedded in paraffin and sectioned into 5 μm-thick slices. Hematoxylin and eosin (H&E) staining was then applied to assess the fundamental morphological features of the tissue. The stained sections were imaged using the Agilent BioTek Cytation C10 Confocal Imaging Reader (Agilent Technologies, Inc., Santa Clara, CA, USA).

### 2.5. Western Blot Analysis

Total proteins were extracted from rat tissues via RIPA containing protease inhibitor PMSF (Beyotime, Shanghai, China). The protein concentration was measured using a BCA assay kit (Beyotime, Shanghai, China). Proteins were separated via SDS-PAGE electrophoresis and transferred to PVDF membranes following the SDS-PAGE protocols. Primary antibodies were incubated with the membrane overnight at 4 °C. The corresponding secondary antibodies were incubated with the membrane for 1.5 h at room temperature. The primary antibodies used were cyclooxygenase-2 (COX-2, CST, 12282; Danvers, MA, USA) diluted at 1:1000 and glyceraldehyde 3-phosphate dehydrogenase (GAPDH, Proteintech, 60004-1-Ig; Wuhan Sanying, Wuhan, China) diluted at 1:50,000. Protein signals were visualized using an enhanced chemiluminescence detection system according to the manufacturer’s instructions (ECL, Thermo Fisher Scientific, Waltham, MA, USA).

### 2.6. Real-Time Quantitative Polymerase Chain Reaction Analysis

Total RNA was extracted from rat tissues using TRIzol. cDNA was synthesized from total RNA using a DNA Removal and Reverse Transcription Kit (RR047A, TAKARA). Real-time PCR was performed using the ChamQ Blue Universal SYBR qPCR Master Mix kitfromVazyme (Q312-02, Nanjing, China) on an Applied Biosystems QuantStudio 6 Pro real-time PCR system to obtain CT values. β-Actin was selected as the reference gene, and the relative expression levels of the target mRNAs were analyzed using the 2^−ΔΔCt^ method. The sequences of the primers used are listed in Table 1.

### 2.7. Statistical Analysis

All results are presented as mean ± SD. All analyses were performed using GraphPad Prism 9.5.1 (GraphPad Software, San Diego, CA, USA). Comparisons between multiple groups were analyzed by one-way ANOVA followed by Dunnett’s post hoc test for multiple comparisons. For comparisons between two groups, Student’s *t*-test was used. When data did not meet the assumptions for parametric tests, nonparametric tests were employed. *p* < 0.05 was considered to indicate statistical significance.

## 3. Results

### 3.1. α-Lack-SPI Has No Effect on Either Body Weight or Liver Weight in Rats Fed with HFD

As shown in Figure 1A, MASLD was induced by HFD for 4 weeks, followed by WT-SPI and α-lack-SPI treatment for another 34 days. Rats in the control group were fed with chow diet during the whole feeding period. The body weight of each group was steadily increased with no difference among the groups at the end of the experiment (Figure 1B and Appendix A). Additionally, compared to the CON group, rats fed with HFD exhibited significantly increased liver weight and hepatosomatic index. While α-lack-SPI treatment significantly reduced the hepatosomatic index, the decrease in liver weight was not statistically significant (Figure 1C,D). In contrast, WT-SPI treatment did not significantly alter these parameters (Appendix A).

### 3.2. α-Lack-SPI Has Positive Effects on Lipid Profiles of Rats with HFD-Induced MASLD

To study the effects of α-lack-SPI on circulating lipid profiles, we assessed the rats’ serum lipid indices, including the levels of TC, LDL-C, HDL-C, and TG (Figure 2A–D). Results demonstrated that serum TC and LDL-C levels were significantly elevated in the HFD group compared to the CON group, whereas HDL-C was significantly reduced. α-lack-SPI administration markedly attenuated HFD-induced elevations in TC and LDL-C, while significantly increasing serum HDL-C levels in HFD-fed rats. Similarly, WT-SPI treatment also significantly reduced serum TC and LDL-C levels in HFD-fed rats, but to a lesser extent compared to α-lack-SPI (Appendix A). As serum TG levels showed no significant difference between HFD and CON groups at endpoint, the effect of α-lack-SPI on TG could not be determined.

### 3.3. α-Lack-SPI Ameliorates Hepatic Steatosis in Rats with HFD-Induced MASLD

In HFD-fed rats, H&E staining of liver sections revealed that hepatocytes exhibited significant swelling and steatosis, which were alleviated following treatment with α-lack-SPI (Figure 3A). Moreover, compared to the CON group, HFD-fed rats exhibited significantly elevated levels of hepatic TG and TC, as well as increased serum levels of ALT and AST (Figure 3B–E). In contrast, α-lack-SPI treatment significantly reduced these elevated levels in HFD-fed rats, resulting in lower hepatic TG and TC (Figure 3B,C) and decreased serum levels of ALT and AST (Figure 3D,E). Besides, results showed that WT-SPI treatment significantly reduced serum AST levels, though not as pronounced as with α-lack-SPI (Appendix A). However, there was no significant difference in serum ALT and hepatic TG levels between the WT-SPI-treated and HFD groups (Appendix A).

### 3.4. α-Lack-SPI Regulates Hepatic Lipid Synthesis and Disposal in Rats with HFD-Induced MASLD

To elucidate the mechanisms by which α-lack-SPI ameliorates hepatic lipid accumulation and dyslipidemia in MASLD rats, we next examined its impact on key genes governing hepatic lipid metabolism in HFD-fed rats by quantitative real-time PCR. Results showed that α-lack-SPI treatment markedly down-regulated hepatic mRNA levels of Srebf1, Acaca, and Fasn up-regulated by HFD feeding, while simultaneously up-regulating Pparα expression compared with the HFD group (Figure 4A–D). Furthermore, α-lack-SPI administration significantly reduced hepatic mRNA levels of Pcsk9 in HFD-fed rats. Simultaneously, α-lack-SPI treatment led to a significant reduction in Hmgcr mRNA levels, even though there was no significant difference in Hmgcr mRNA levels between the HFD and CON groups (Figure 4E,F).

### 3.5. α-Lack-SPI Inhibits Hepatic Inflammation in Rats with HFD-Induced MASLD

High-fat diet feeding can trigger a chronic low-grade inflammatory state through multiple pathways, which is closely intertwined with the progression of MASLD. To determine the effects of α-lack-SPI on hepatic inflammation, we measured the hepatic mRNA levels of five transcripts, including Tnf-α, Il-1β, Il-6, Ccl2, and Nlrp3 levels, all of which were markedly elevated in the HFD group compared to the CON group. These elevations were significantly reversed by α-lack-SPI treatment (Figure 5A–E). Concurrently, COX-2 protein expression was significantly up-regulated in the livers of HFD-fed rats but was restored to near normal by α-lack-SPI administration (Figure 5F,G). However, these effects were not observed in HFD-fed rats treated with WT-SPI (Appendix A–D). Together, these findings demonstrated that α-lack-SPI markedly attenuated HFD-induced hepatic inflammation in MASLD rats.

## 4. Discussion

Our study observed that therapeutic treatment with α-lack-SPI significantly reduced hepatic lipid accumulation and improved systemic lipid metabolism, and ameliorated steatohepatitis-associated hepatic injury. α-lack-SPI alleviated HFD-induced MASLD in rats, probably by ameliorating hepatic lipogenesis, regulating cholesterol homeostasis, and reducing inflammation.

The development of MASLD is intricately linked to lipid metabolism, especially the metabolism of triglycerides and cholesterol [30]. The initial manifestation of MASLD is the excessive accumulation of lipids within hepatocytes [31]. We observed significant improvement in the triglyceride and cholesterol content of the liver tissue of the HFD-fed rats treated with α-lack-SPI. Hepatic lipid accumulation is a direct consequence of DNL, which involves several rate-limiting enzymes that have emerged as prime therapeutic targets for MASLD [32,33,34]. Our study demonstrated that α-lack-SPI treatment significantly down-regulated hepatic expressions of key genes involved in DNL, including Srebf1, Acaca, and Fasn, in HFD-fed rats, thereby attenuating lipid synthesis. In addition, PPARα is a critical transcription factor in the liver that facilitates the β-oxidation of excess fatty acids to produce adenosine triphosphate (ATP) [35,36]. Treatment with a PPARα agonist enhanced the mitochondrial density per liver area, emerging as a potential target to counter MASLD or to protect against the progression toward MASH [37]. In α-lack-SPI-treated rats, the upregulation of PPARα mRNA in the liver suggested increased PPARα activation, which likely accelerated hepatic fatty-acid β-oxidation and reduced lipid accumulation.

Elevated cholesterol profiles were often observed in patients with MASLD [12,13,38]. Studies have shown that as MASLD progresses to more severe stages such as MASH, fibrosis, and cirrhosis, cholesterol synthesis tends to increase [39]. For decades, the serum cholesterol-lowering potency of soy constituents has been widely recognized [40,41,42]. Growing evidence now centers on the divergent hypolipidemic efficacy of soy protein isolates with distinct subunit profiles. Specifically, the cholesterol-lowering capacity of soy protein isolate is differentially contributed by distinct subunits: the α′ subunit of 7S and the A1aB1b and A2B1a subunits of 11S [43]. The 7S α′ subunit plays a pivotal role in cholesterol homeostasis by up-regulating hepatic LDL receptors, thereby reducing circulating cholesterol levels [28,29,44]. Additional evidence indicates that deletion of the α subunit elicits a compensatory increase in α′ subunit in SPI [23]. Consistent with these findings, α-lack-SPI markedly down-regulated hepatic Pcsk9 mRNA expression in HFD-fed rats. The PCSK9 protein was characterized as the ninth member of the proprotein convertase subfamily [28]. Circulating PCSK9 elevations drive LDL receptor degradation on the hepatic surface, reduce LDL-C uptake, and eventually elevate plasma LDL-C concentrations [45]. Thus, our findings suggested that treatment with α-lack-SPI maintained cholesterol homeostasis in HFD-fed rats by attenuating hepatic PCSK9 synthesis, thereby preventing LDL-R degradation.

Lipid accumulation in the liver further induces lipotoxicity, which exacerbates the occurrence of hepatic inflammation. Diverse lipid species, sensed by pattern-recognition receptors (PRRs) either extracellularly or intracellularly, induce the production of cytokines within hepatocytes or immune cells [46]. NLRP3 is an intracellular pattern-recognition receptor (PRR) that can be activated by microbial signals, damage-associated molecular patterns (DAMPs) such as ATP, and certain lipid species [47]. Earlier studies have shown that saturated fatty acids in Western diets can activate the NLRP3 inflammasome in hematopoietic cells [48]. As demonstrated in our study, in HFD-fed rats, the significant upregulation of hepatic Nlrp3 and Il-1β mRNA levels reflected intensified hepatic inflammation, which was markedly attenuated following α-lack-SPI treatment. Additionally, in the livers of HFD-fed rats, we observed significant upregulation of mRNA expression levels of pro-inflammatory cytokines such as Tnf-α and Il-6, which are not only key drivers of MASLD progression [49] but also act in concert with the chemokine CCL2 to stimulate macrophage infiltration [50,51,52]. α-lack-SPI may alleviate MASLD in HFD-fed rats by reducing the formation of hepatic inflammasomes and macrophage infiltration. Moreover, COX-2 is involved in mediating inflammatory responses and is highly expressed in the livers of patients with MASLD [53,54]. Many clinical drugs, such as celecoxib and valdecoxib (COX-2 inhibitors), have been used to alleviate hepatic steatosis or lipid accumulation in MASLD models [55,56]. Deletion of COX-2 has been shown to mitigate inflammation in the liver and adipose tissue induced by HFD [57]. Our results demonstrated that α-lack-SPI administration significantly reduced hepatic COX-2 protein expression in HFD-fed rats, and its beneficial effects on MASLD may also be partly attributed to its COX-2 targeting inhibitory effects.

Soy protein is a vital plant-based protein source in the diet, with nutritional value comparable to that of animal protein, and is considered a full-value protein [58,59]. The digestion of soy protein in the gastrointestinal tract yields a variety of peptides, which are absorbed by the intestines into the blood circulation to exert physiological effects in target tissues [60,61], and are considered one of the sources of its nutritional attributes [62,63,64]. Three peptides derived from 7S globulin (KNPQLR, EITPEKNPQLR, and RKQEEDEDEEQQRE) can inhibit FAS [65]. In a previous study, YVVNPDNDEN and YVVNPDNNEN were peptides that act as competitive inhibitors of HMG-CoA-R, exhibiting statin-like behavior, while the latter one also down-regulated the protein level of PCSK9, a key regulator of LDL-R degradation [66]. Additionally, lunasin is a 43-amino-acid soybean-derived peptide, which displayed anti-inflammatory effects by reducing the generation of COX-2 and the production of TNF-α, interleukin-6, and other pro-inflammatory cytokines [67]. Therefore, the observed alleviation of MASLD by α-lack-SPI was likely not due to intact proteins, but rather to peptides derived from the proteins by gastrointestinal digestion.

The specific subunit composition of SPI, particularly the presence or absence of certain subunits, significantly influences its functional properties and nutritional value, which are crucial for its applications. As previous studies have indicated, the absence of the α subunit leads to a compensatory increase in the α′ subunit, which is a primary source of the nutritional value of SPI [23,25]. In the present study, we found the advantages of α-lack-SPI in regulating hepatic lipid metabolism and attenuating hepatic inflammation. Specifically, α-lack-SPI treatment resulted in significantly lower levels of serum TC, LDL-C, ALT, AST, and liver TG, as well as markedly reduced mRNA levels of hepatic Tnf-α, Il-1β, Il-6, Ccl2, and Nlrp3, and decreased protein expression of hepatic COX-2 in HFD-fed rats, as shown in the Appendix A. These results indicated that α-lack-SPI possessed enhanced lipid-lowering properties and highlighted its anti-inflammatory advantages. The superior efficacy of α-lack-SPI in mitigating MASLD in HFD-fed rats may be attributed to its unique subunit composition.

To our knowledge, this is the first study to investigate the effects of α-subunit-deficient soy protein isolate on MASLD in HFD-fed rats. We found that α-lack-SPI alleviated MASLD in HFD-fed rats by reducing hepatic lipid accumulation, maintaining cholesterol homeostasis, and mitigating inflammation caused by lipotoxicity. This study did not identify which specific active compound of the α-lack-SPI is responsible for these effects, which requires further investigation.

## 5. Conclusions

This study indicates that α-lack-SPI treatment significantly ameliorates hepatic steatosis and steatohepatitis by regulating the expression of genes of lipid metabolism and inflammation in the liver. The study offers a novel perspective for the nutritional management of MASLD, with α-lack-SPI emerging as a promising functional food component for its amelioration.

## Figures and Tables

**Figure 1 nutrients-17-02918-f001:**
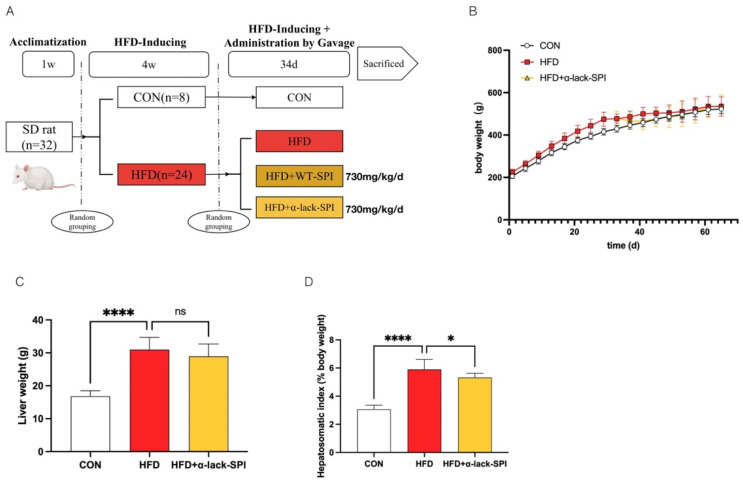
Effects of α-lack-SPI treatment on basic physiological parameters of rats with HFD-induced MASLD. (**A**) Animal experimentation process, (**B**) body weight, (**C**) liver weight, (**D**) hepatosomatic index. The data are presented as means ± SD (*n* = 8). Compared with the HFD group, * *p* < 0.05, **** *p* < 0.0001; ns, not significant.

**Figure 2 nutrients-17-02918-f002:**
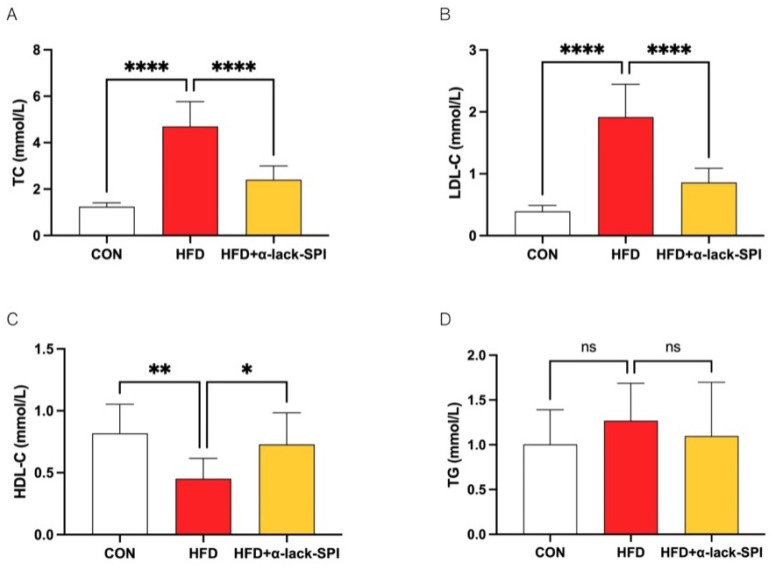
Effects of α-lack-SPI on serum (**A**) TC, (**B**) LDL-C, (**C**) HDL-C, (**D**) TG of rats with HFD-induced MASLD. The data are presented as means ± SD (*n* = 8). Compared with the HFD group, * *p* < 0.05, ** *p* < 0.01, **** *p* < 0.0001; ns, not significant.

**Figure 3 nutrients-17-02918-f003:**
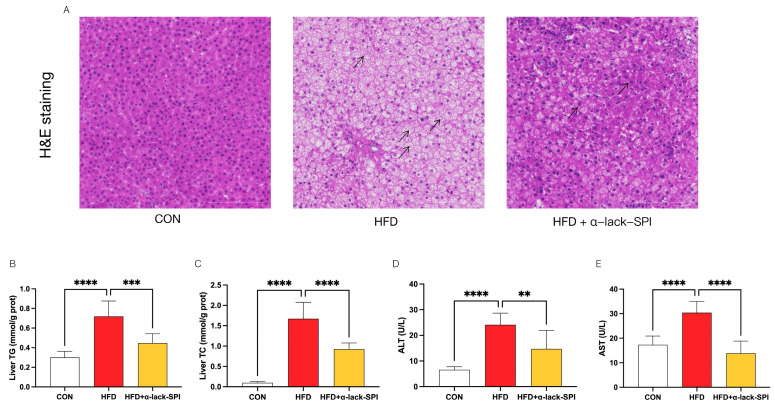
Effects of α-lack-SPI on hepatic steatosis and damage in rats with HFD-induced MASLD. (**A**) Histopathological analysis of the liver by H&E staining (200×; scale bar: 200 µm). Arrows highlight the differences in hepatocyte swelling and steatosis between the HFD and α-lack-SPI treatment groups. Liver (**B**) TG and (**C**) TC levels. Serum (**D**) ALT and (**E**) AST activities. The data are presented as means ± SD (*n* = 8). Compared with the HFD group, ** *p* < 0.01, *** *p* < 0.001, **** *p* < 0.0001.

**Figure 4 nutrients-17-02918-f004:**
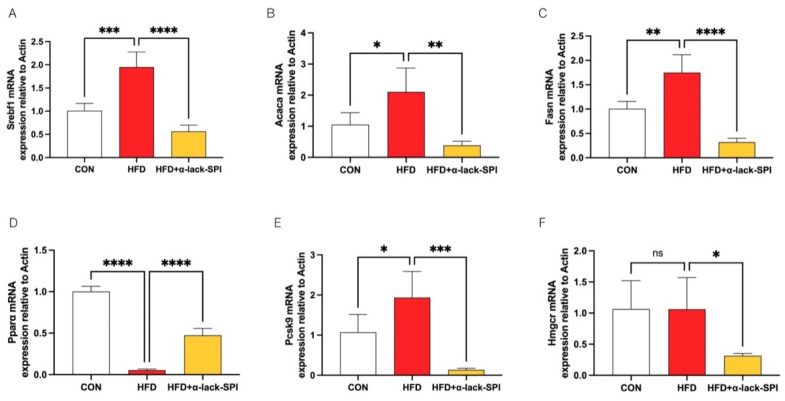
Effects of α-lack-SPI on the mRNA expressions of hepatic lipid metabolism-related genes in rats with HFD-induced MASLD. (**A**) Srebf1, (**B**) Acaca, (**C**) Fasn, (**D**) Pparα, (**E**) Pcsk9, (**F**) Hmgcr. Values were normalized to the reference gene Actb. The data are presented as means ± SD (*n* = 4). Compared with the HFD group, * *p* < 0.05, ** *p* < 0.01, *** *p* < 0.001, **** *p* < 0.0001; ns, not significant.

**Figure 5 nutrients-17-02918-f005:**
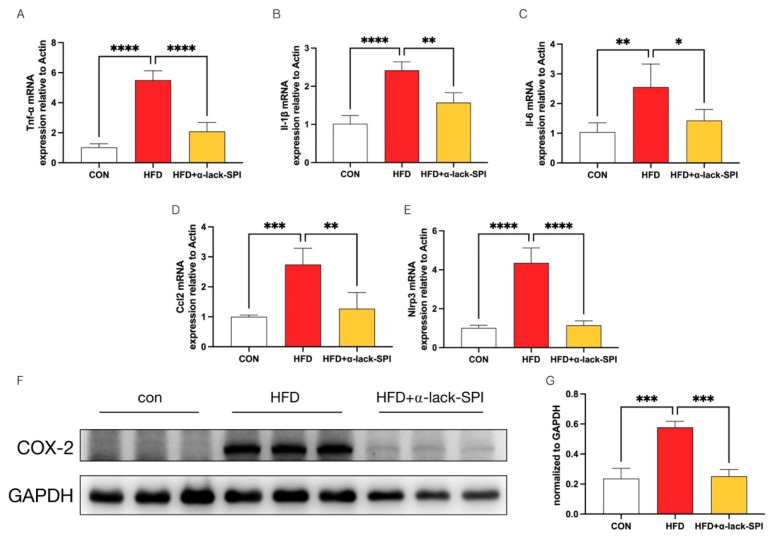
Effects of α-lack-SPI on hepatic inflammatory mRNA and protein expressions in rats with HFD-induced MASLD. Hepatic mRNA levels of Tnf-α, Il-1β, Il-6, Ccl2, Nlrp3 (**A**–**E**), and hepatic COX-2 protein levels (**F**,**G**) were measured and normalized to GAPDH (*n* = 3). Values were normalized to the reference gene Actb. The data are presented as means ± SD (*n* = 4). Compared with the HFD group, * *p* < 0.05, ** *p* < 0.01, *** *p* < 0.001, **** *p* < 0.0001.

**Table 1 nutrients-17-02918-t001:** The sequence of RT-qPCR primer.

Gene	Species	Forward Primer	Reverse Primer
Srebf1	rat	TCTTGACCGACATCGAAGACAT	CTGTCTCACCCCCAGCATAG
Acaca	rat	GCGGCTCTGGAGGTATATGT	GGGATGTTCCCTCTGTTTGGA
Fasn	rat	GCCTAACACCTCTGTGCAGT	GGCAATACCCGTTCCCTGAA
Pcsk9	rat	TTTCCACAGACAGGCGAGC	CCTTGACAGTTGAGCACACG
Pparα	rat	CTGAACATTGGCGTTCGCAG	TTCAGTCTTGGCTCGCCTC
Hmgcr	rat	TGCAGAGCGATCAGTCTTGG	AATCTGCTCGTGCTGTCGAA
Nlrp3	rat	CTGCAGAGCCTACAGTTGGG	ACCCTACACTAAAAGCGCCC
Tnf-α	rat	GATCGGTCCCAACAAGGAGG	CTTGGTGGTTTGCTACGACG
Il-1β	rat	GGCTTCCTTGTGCAAGTGTC	AGTCAAGGGCTTGGAAGCAA
Il-6	rat	GCCCACCAGGAACGAAAGTC	ACTGGCTGGAAGTCTCTTGCG
Ccl2	rat	TTAATGCCCCACTCACCTGC	GAGCTTGGTGACAAATACTACAGC
Actb	rat	TGCTATGTTGCCCTAGACTTCG	GTTGGCATAGAGGTCTTTACGG

Srebf1: Sterol regulatory element-binding transcription factor 1; Acaca: Acetyl-CoA carboxylase; Fasn: Fatty acid synthase; Pcsk9: Proprotein convertase subtilisin/kexin type 9; Pparα: Peroxisome proliferator-activated receptor alpha; Hmgcr: 3-Hydroxy-3-methylglutaryl-CoA reductase; Nlrp3: Nucleotide-binding domain, leucine-rich repeat-containing family, pyrin domain-containing 3; Tnf-α: Tumor necrosis factor alpha; Il-1β: Interleukin-1 beta; Il6: Interleukin-6; Ccl2: Chemokine (C-C motif) ligand 2; Actb: Actin, beta.

## Data Availability

All data that support the findings of this study are available from the authors upon reasonable request due to privacy reasons.

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
