# Peer review of "α-Lack-SPI Alleviates MASLD in Rats via Regulating Hepatic Lipid Accumulation and Inflammation"

_nutrients, 2025, doi:10.3390/nu17182918_

Round 1
Reviewer 1 Report
Comments and Suggestions for Authors
The manuscript describes the potential effects of alpha-lack-SPI treatment on liver metabolism in rats fed high fat diets to attenuate MASLD. The manuscript is well written and clearly demonstrates the specific effect of alpha unit-deficient SPI compared to SPI alone in animals on HFD. There are minor typos highlighted in red in the attached file. For better clarity, it would be helpful if you could increase the magnification of Figure 3A to more clearly visualize the effect of steatosis.

Author Response
Comments 1: The manuscript describes the potential effects of alpha-lack-SPI treatment on liver metabolism in rats fed high fat diets to attenuate MASLD. The manuscript is well written and clearly demonstrates the specific effect of alpha unit-deficient SPI compared to SPI alone in animals on HFD. There are minor typos highlighted in red in the attached file. For better clarity, it would be helpful if you could increase the magnification of Figure 3A to more clearly visualize the effect of steatosis.
Response 1: Thank you for your positive feedback and constructive suggestions! We have carefully reviewed the highlighted typos and made the necessary corrections. All changes have been highlighted in the revised manuscript. (lines 41, 52, 55, 114-116, 280, 327)
Regarding your suggestion to increase the magnification of Figure 3A, we have enlarged the image size to provide a clearer view of the histological details. Additionally, we have used arrows to highlight the differences in steatosis between the HFD and HFD+α-lack-SPI groups. The updated figure now clearly visualizes the effect of steatosis, making it easier to compare the two groups.
We appreciate your attention to detail and believe that these changes have improved the clarity and presentation of our results.

Reviewer 2 Report
Comments and Suggestions for Authors
This study investigates the potential of an a-lack-SPI to improve MASLD in an experimental model. The work is well-structured and the findings are promising. The data convincingly show improvements in hepatic lipid profiles and a reduction in inflammatory markers.
I have the following comments:
-
You suggest that the unique composition of a-lack-SPI, particularly the compensatory increase in the a' subunit, drives its efficacy. Could you provide data on the actual subunit/peptide composition of the a-lack-SPI used?
-
The discussion mentions that peptides derived from GI digestion are likely responsible for the observed effects. Was any analysis conducted to identify these bioactive peptides in vivo? If not, this could be a promising field for further research,
-
In Figure 5, the sample size for the Western Blot appears to be n=3, while other experiments use n=4 or n=8. Please clarify this discrepancy.
Author Response
Comments 1: You suggest that the unique composition of a-lack-SPI, particularly the compensatory increase in the a' subunit, drives its efficacy. Could you provide data on the actual subunit/peptide composition of the a-lack-SPI used?
Response: Thank you for your insightful comment! We have now provided detailed data on the subunit composition of both WT-SPI and α-lack-SPI in the supplementary materials (Supplementary Figure S4). As discussed in the manuscript, the absence of the α subunit in α-lacking-SPI appears to result in a compensatory increase in the α' subunit. Although we have not performed quantitative analysis, previous studies have already indicated that the absence of the α subunit can lead to a compensatory increase in the α' subunit [1, 2]. We believe that the new data provided, in conjunction with the established literature, robustly supports our interpretation and hope that this addresses your question properly.
1. Liu, S.S.; Luo, T.T.; Song, Y.R.; Ren, H.B.; Qiu, Z.D.; Ma, C.X.; Tian, Y.S.; Wu, Q.; Wang, F.; Krishnan, H.B.; et al. Hypocholesterolemic effects of soy protein isolates from soybeans differing in 7S and 11S globulin subunits vary in rats fed a high cholesterol diet. J Funct Foods 2022, 99, 105347.
2. Song, B.; Qiu, Z.; Li, M.; Luo, T.; Wu, Q.; Krishnan, H.B.; Wu, J.; Xu, P.; Zhang, S.; Liu, S. Breeding of ‘DND358’: A new soybean cultivar for processing soy protein isolate with a hypocholesterolemic effect similar to that of fenofibrate. J Funct Foods 2022, 90, 104979.
Comments 2: The discussion mentions that peptides derived from GI digestion are likely responsible for the observed effects. Was any analysis conducted to identify these bioactive peptides in vivo? If not, this could be a promising field for further research,
Response: Thank you for your professional feedback! While our study demonstrates the potential effects of α-lack-SPI treatment on rats with HFD-Induced MASLD, we have not yet conducted in vivo analysis to identify the bioactive peptides derived from gastrointestinal digestion. As you pointed out, we have discussed the potential effects of these peptides in the discussion section. We have highlighted the metabolic regulatory capabilities of bioactive peptides derived from soy protein isolate and their potential contribution to the observed effects in our study.
We agree that further investigation into these bioactive peptides is a promising area for future research. In our subsequent studies, we plan to focus on identifying and characterizing these peptides in more detail, which will provide a deeper understanding of their mechanisms of action and their potential therapeutic applications.
We appreciate your suggestion and will prioritize this area in our future research endeavors.
Comments 3: In Figure 5, the sample size for the Western Blot appears to be n=3, while other experiments use n=4 or n=8. Please clarify this discrepancy
Response: Thank you for your careful review! We appreciate the opportunity to clarify our experimental design, which indeed involved different sample sizes for different techniques based on their distinct technical requirements and established practices in the field. Below is a detailed explanation:
For the measurement of biochemical parameters, we were able to analyze samples from all n=8 individual animals per group, as these assays are highly quantitative, and require minimal sample material.
For the RT-qPCR experiments, we analyzed n=4 biological replicates per group. This sample size was determined to be sufficient to achieve statistical power based on expected effect sizes and variability commonly observed in gene expression studies. The samples were obtained from a randomly selected subset of n=4 animals per group. We appreciate it that you raise this key point and will be careful in analyzing all samples in upcoming RT-qPCR analysis.
In western blotting, ‘n=3’is the number of representative biological replicates shown in the manuscript. Actually, we did the analysis of all the samples of each group.
Thank you again for raising this point.

Reviewer 3 Report
Comments and Suggestions for Authors
In the current study, the authors investigated the therapeutic potential and underlying mechanisms of a specialized soy protein isolate α-lack-SPI in a high-fat, high-cholesterol diet-induced MASLD rat model, focusing on its effects on regulating hepatic lipid metabolism and inhibiting inflammation. Their findings indicates that α-lack-SPI may serve as a promising nutritional intervention for MASLD management.
Some suggestions:
- Please add for which year is valid the statistical data presented at lines 36-38.
2.Lines 112-113, you wrote “Suspensions of each compound were prepared by dissolving them in 5 mL of 0.5% CMC-Na solution”. If you obtained a suspension is not correct to say that you dissolved the compound. Please correct.
- Lines 114-115, you wrote “Rats in the CON and HFD groups received an equal volume of the vehicle via gavage”. Please specify about what volume you are talking about.
- lines 117-118: Please add how the rats were sacrificed and what do you mean by “other samples for further analysis”.
5.Point 2.3, lines 120-127: What devices were used to determine the biochemical parameters. Please add.
6.Point 2.4. Add please the type of the microscope used to perform the histopathological analysis.
Author Response
Comments 1: Please add for which year is valid the statistical data presented at lines 36-38.
Response: Thank you for your suggestion! We have revised the sentence to specify the time period based on the data from the relevant studies.
The updated sentence now reads: "MASLD has been estimated to affect 30% of the adult population worldwide during 1999-2019, and is becoming an important public health problem all over the world." (lines 37-39)
Comments 2: Lines 112-113, you wrote “Suspensions of each compound were prepared by dissolving them in 5 mL of 0.5% CMC-Na solution”. If you obtained a suspension is not correct to say that you dissolved the compound. Please correct.
Response: Thank you very much for your valuable feedback! We have revised the sentence to more accurately describe the preparation of the suspensions. During the revision process, we also discovered that the concentration of carboxymethyl cellulose sodium (CMC-Na) was incorrectly stated as 0.5% in the initial manuscript. We have corrected this to the accurate concentration of 0.05%.
The revised sentence now reads: “Suspensions of each compound were prepared by dispersing them in 5 mL of 0.05% carboxymethyl cellulose sodium solution.” (lines 114-116)
Comments 3: Lines 114-115, you wrote “Rats in the CON and HFD groups received an equal volume of the vehicle via gavage”. Please specify about what volume you are talking about.
Response: Thank you for pointing this out. We have revised the sentence to specify the volume of the vehicle administered to the rats.
The updated sentence now reads: “Rats in the CON and HFD groups received an equivalent volume of the vehicle alone, also based on their body weight to the treatment group.” (lines 116-118)
Comments 4: lines 117-118: Please add how the rats were sacrificed and what do you mean by “other samples for further analysis”.
Response: Thank you for your valuable feedback! We have revised the sentence to more clearly describe the method of sacrificing the rats and the subsequent collection of samples.
The updated sentence now reads: “The treatment lasted for a total of 34 days, after which euthanasia was performed on rats through cardiac puncture under isoflurane anesthesia. Blood and liver tissue were then collected for further analysis.” (lines 120-122)
Comments 5: Point 2.3, lines 120-127: What devices were used to determine the biochemical parameters. Please add.
Response: Thank you for your suggestion. We have added the specific model of the microplate reader used for the measurements to the methods section.
The sentence now reads: “Measurements were performed using a Spark 10M multimode microplate reader (Tecan Trading AG, Männedorf, Switzerland).” (lines 131-132)
Comments 6: Point 2.4. Add please the type of the microscope used to perform the histopathological analysis.
Response: Thank you for your careful review! We have incorporated this information into the manuscript.
The sentence now reads: “The stained sections were imaged using the Agilent BioTek Cytation C10 Confocal Imaging Reader (Agilent Technologies, Inc., Santa Clara, CA, USA).” (lines 138-139)
